

# Snow gliding and glide snow avalanches: recent outcomes from two experimental test sites in Aosta Valley (NW Italian Alps)

Margherita Maggioni[1,2], Danilo Godone[2,3], Barbara Frigo[4], Michele Freppaz[1,2]

5   [1]DISAFA, University of Torino, Grugliasco (TO), 10095, Italy
[2]NatRisk, University of Torino, Grugliasco (TO), 10095, Italy
[3]IRPI, National Research Council, Torino, 10135, Italy
[4]DISEG, Politecnico of Torino, 10129, Italy

*Correspondence to*: Margherita Maggioni (margherita.maggioni@unito.it)

**Abstract**. Snow gliding and glide snow avalanches are gaining importance among scientists as climate change might induce conditions favourable to those phenomena. Our aim is to analyse such processes with a particular

focus on the potential driving factors associated to the soil conditions. We equipped two experimental test sites in Aosta Valley Region (NW-Italy) with glide-snow shoes, temperature and volumetric liquid water content (VLWC) sensors in the soil and in the basal snowpack layer; snow and weather parameters were also collected by automatic weather stations and in manual snow measuring sites.

In the two monitoring seasons 2013-14 and 2014-15 we registered 9 glide snow avalanches: 2 cold and 7 warm-

20 temperature events, which were characterized by different snow and soil parameters. In the only warm glide snow avalanche event, which presented a continuous gliding before, the daily glide rate showed a significant exponential relationship with the soil VLWC. We also found, though without a general trend, that gliding and non-gliding periods (either considering warm and cold periods separately or together) were characterized by significantly different predisposing factors.

This study contributes to assess the importance of soil VLWC, which seems to be one of the most important driving factors for gliding processes. Therefore, it supports the need, already suggested by other scientists, of analysing such processes with an interdisciplinary approach which integrates snow and soil sciences.

## 1 Introduction

Snow gliding processes have recently gained importance among scientists as climate change might influence the

30 snow cover and the related processes, leading to a reduction of dry snowpacks and an increase of wet ones (Castebrunet et al., 2014). Already in the past, scientists have studied snow gliding and glide snow avalanches - see reviewing papers by Holler (2014) and Ancey (2015), but current and future climate change might generate scenarios prone to such processes, therefore they are gaining again importance. In fact, raise of the air temperature and of the snowfall limit (Hartman et al., 2013) may influence the snow cover, which might become

denser and wetter, also due to more frequent rain on snow events, in particular above 2000 m asl (Morán-Tejeda et al., 2018).



A deep knowledge of the most important driving factors of such phenomena deserve attention, in order to be able to eventually predict and manage them in the optimum way. Already In der Gand and Zupancic (1966) states that snow gliding occurs on a smooth ground surface, a lowermost layer of wet snow and a temperature at the snow-ground surface of 0 °C. Based on different driving factors, some scientists elaborated models to predict snow gliding distances (e.g. Leitinger et al., 2008) or to identify areas prone to snow gliding (e.g. Maggioni et al., 2016).

The approach commonly used for studying snow gliding and glide snow avalanches starts from the distinction between cold and warm temperature events. This distinction is based on the origin of liquid water at the snow-soil interface: in a cold-temperature event, the necessary wet snow-soil interface originates either from snow melting at basal layers of the snowpack or from suction; in a warm-temperature event, the water originates from melting processes at the snow surface, percolates through the snowpack and ponds at the snow-soil interface. The most recent studies by Ceaglio et al. (2017) and Fromm et al. (2018) underline the importance of soil moisture as a driving factor for snow gliding. In particular, Ceaglio et al. (2017) found a strong relation between glide snow rate and moisture at the snow-soil interface for cold temperature events, with a possible contribution also of soil liquefaction, while Fromm et al. (2018) found that soil moisture and temperature had a significant influence on snow gliding in both warm and cold periods.

As in Ceaglio et al. (2017) and Fromm et al. (2018), this work, using an experimental approach, starts from data gathered in two test sites and aim at giving a contribution to the on-going scientific discussion on the most important driving factors for snow gliding and glide snow avalanches. Our goal is to quantify the linkages between soil water content and snow gliding processes and to assess the factors predisposing the glide avalanches release, under different snowpack conditions (warm vs cold events).

## 2 Data and methods

### 2.1 Study area

The study area is located within the MonterosaSki resort in the NE of the Aosta Valley Region (NW Italian Alps), in the Mont Rose Massif, close to the LTER site Istituto Mosso (https://deims.org/17210eba-d832-4759-89fa-9ff127cbdf6e). From the data taken at the weather station D'Ejola, at 1850 m asl, the long-term mean precipitation (including the snow water equivalent) is 1111 mm yr$^{-1}$ (years 1927-2017), the mean annual air temperature is 4.2 °C (years 1971-2017) and the average cumulative snowfall is 398 cm (years 1981-2010).

Two test sites were equipped in the study area for measuring snow gliding and snow and soil properties. The site "Pista Nera" (PN) is a slope at about 2230 m asl with an average inclination of 40° with an ESE aspect. The soil is classified as Cambisol and is characterized by a Liquid Limit (LL) in the topsoil (0-10 cm depth) and in the underlying soil horizon (10-20 cm depth) equal to 53 % and 47 %, respectively. The slope is covered by a mountain grassland (*Festuca scabriculmis*) with sparsely low dwarf shrubs (*Juniperus nana*, *Calluna vulgaris* and *Rhododendron ferrugineus*). The site "Sant'Anna" (SA) is a slope at about 2120 m asl with an average inclination of 36° with an E aspect. The soil is classified as Regosol, with a Liquid Limit of 84 % and 76 % in the topsoil and in the underlying soil horizon, respectively. The dominant plant species are *Festuca rubra s.l.* and *Agrostis tenuis*. The glide snow avalanches releasing from the PN site can reach the ski run below (Fig. 1 and 2) and therefore needs a careful monitoring.





### 2.2 Data collection

The data were collected during the hydrological years 2013-2014 and 2014-2015. Snow and meteorological parameters (snow depth, precipitation, air temperature, wind speed and direction) were provided by the automatic weather station (AWS) "Gressoney-S.J. – Weissmatten", which is operational since 2002 and placed

12 km further south from the study site at 2038 m asl. The new snow amount in 24h was gathered from the manual snow measuring site "Gressoney-L.T. – Lake Gabiet" (2370 m asl) 4 km east from the study sites.

To describe the physical properties of the snowpack, we used the weekly snow profiles made at the manual snow measuring site Sant'Anna, very close (less than 1 km) to the study sites, by the *Corpo Forestale Valdostano* (Forestry Office). Observations were performed according to Fierz et al. (2009).

In the two test sites, instrumentation was installed for measuring snow gliding and snow and soil properties (Fig. 3). In each test site, two glide-snow shoes, connected to potentiometers (Sommer®), were placed within the area where glide-cracks were observed in the past. In addition to the glide-snow shoes, temperature sensors (Campbell - 107 Temperature Probe) and volumetric liquid water content probes (Campbell-CS616 - Water Content Reflectometers WCR) were placed at the snow-soil interface and at two different soil depths (5 cm and

15 cm). The scheme of the instrumental set-up for both sites is given in Fig. 3. Moreover, a web-cam was continuously monitoring the possible glide cracks formation and evolution in the PN site.

### 2.3 Data analysis

During the two monitoring seasons, we selected different periods for the analyses of the data registered by each single glide shoe in the two test sites (Tab. 1). In fact, the four glide shoes showed different patterns (see later in

section 3). Moreover, the periods for the analyses were also chosen according to the specific snow conditions of the sites, which influenced the significance of the registered temperature and volumetric liquid content (VLWC) parameters. For example, after a glide avalanche event, identified as a large sudden glide shoe movement and/or from field survey, the instrumentation was reset but the site remained with no snow until the next snowfall, therefore this period could not be considered for analysis.

We considered daily values, which were obtained by averaging the 30 minutes values for all parameters, except for the daily glide-snow rate, which was calculated as the difference of the cumulative gliding at 23:30 h between two consecutive days.

We performed statistical analyses to investigate the difference between cold and warm gliding events in terms of driving factors (snow and soil temperature and VLWC, air temperature, snow height, new snow in 24h). We

assume that the distinction between a cold and a warm temperature event is related to the origin of the liquid water at the snow-soil interface as in Ceaglio et al. (2017).

Concerning glide-snow avalanche events, we explore possible differences between driving factors measured at the moment of the events (semi-hourly data) for warm and cold events, applying the Wilcoxon-Mann-Whitney test.

We executed univariate (Mann-Witney U-test) and multivariate (Classification Trees) statistical analyses to explore differences between periods of gliding (identified as those days with a daily glide rate greater than 0.5 cm $d^{-1}$ for at least one glide shoe) and periods of no gliding. Initially, we considered the whole dataset at once and then we classified into cold- and warm-temperature periods in the two test sites, in order to investigate the potential differences in the driving factors.



We also checked potential relationships between glide rate and soil VLWC.

Beside the above mentioned quantitative analyses, we also made a qualitative description of the two monitoring seasons, trying to explain some patterns found in the registered values for specific days.

### 3 Results and discussion

*Predisposing factor for glide avalanche events*

Figures 4-7 report the pattern of the registered variables along the two monitoring seasons in the two test sites. In total, we registered 9 glide snow avalanches. Other events occurred in the first season, but were only observed through field surveys without measurements by the glide-snow shoes. Among the measured events, 2 have been classified as cold events and 7 as warm events (among which 1 was related to a rain-on-snow episode). Often the

10 events occurred suddenly without an appreciable snow gliding before.

The two cold glide-snow avalanche events were recorded in late fall 2013 in both sites, while the warm events were recorded both in fall and spring. In literature (e.g. Dreier et al., 2016), cold events were usually associated to the beginning of the winter season, while warm ones only to the end. But the beginning of season 2014-15 was meteorologically an exception. In fact, November 2014 was the second warmest (air temperature 3.1 °C

above the average) and the rainiest (378 % more than the average) November of the previous 57 years in NW Italian Regions (ARPAP, 2015). The conditions were not typically winter; the first available snow profile (December 17th) described a wet snowpack with melt forms in the basal layer. We classified the registered gliding events occurred in November 2014 as warm; on November 30th a glide avalanche due to a ROS event occurred. For such events VLWC was around 35 % in SA, while in PN it started around 15 % (November 5th)

and raised up to 22 % on November 30th. The soil was not frozen in both sites. In comparison to 2013-14 the soil showed the same values of VLWC in SA, while in PN the soil was wetter in November 2014 than in November 2013. Soil in SA was in general wetter than in PN in both seasons.

Analysing the patterns of air temperature and snow height, we could appreciate the fact that the warm temperature events often occurred after a snowfall followed by a sharp increase in air temperature, resulting in a

25 decrease in the snow depth (Fig. 4-7). For warm events this might describe the situation in which snow melting occurred at the snow surface with percolating water through the snowpack down to the snow-soil interface with a lowering of the basal friction. The same occurred for the cold events in November 2013 at both sites, but with a less sharp decrease in the snow depth.

Comparing SA and PN after the first snowfall at the end of November 2013, it interesting to notice how this

snowfall caused a snow gliding phenomenon in PN site earlier than in SA. In PN a glide avalanche occurred on the 21st, while in SA on the 24th. From the data, soil temperature in PN (2.9 °C) was 1 °C warmer than in SA (1.8 °C), while soil moisture was about the half in PN (19 %) than in SA (35 %). Probably, the reason of the earlier movement in PN than in SA lies in the average inclination of the sites, which is 40° for PN and 36° for SA, and in the vegetation characteristics. Vegetation in PN is more favourable than that in SA to gliding: though the

presence of sparse lignified shrubs, PN presents longer grass, not grazed, while SA is usually grazed by dairy cows. However, despite some differences between the two sites, their characteristics make them to belong both to classes which present a low basal friction, favourable to snow gliding (Feistl et al., 2014).

The complexity of the experimental test sites, with couples of glide shoes which behaved differently and not like replications, could gave us the possibility of trying to explain some differences between the glide shoes.



For example, at the beginning of season 2013-14, in PN a glide avalanche occurred on November 21[st] involving glide shoe S2 in its movement (Fig. 4). This avalanche occurred after the first consistent snowfall of more than 1 m of new snow (Fig. 4). After this first event, no snowfall occurred until December 19[th] when 30 cm of new snow fell again. Glide shoe S2, which was reset on November 27[th], did not move anymore until the last small and fast movement in April 2014. The data of soil temperature and moisture showed that the soil remained frozen until January 14[th]. The instruments were closer to S2 than S1, representing then the situation of no (or little) snow at ground between the event on November 24[th] (which denudated the soil) and January 14[th]. Therefore, S2 did not move because one of the necessary conditions for gliding was not present (snow-soil interface at 0 °C). Instead S1, which was not involved in the glide avalanche of November 24[th], remained below 1 m of snow plus 30 cm from the snowfall on December 19[th], when it started moving again continuously. At the location of S1 the snow at ground was more than 1 m and possibly the soil was not frozen, favouring a continuous and gradual snow gliding, as registered by S1.

A simple descriptive statistics of the variables registered at the moment of the glide avalanche events (box-plots in Fig. 8) showed that the cold glide snow avalanche events were characterized, in average, by: i) a slightly higher soil VLWC (26.9 % and 25.3 %) than in case of warm events (25.6 % and 24.9 %) at 5 cm and 15 cm depths, respectively; ii) a lower VLWC (0.6 %) in the snowpack basal layer than in case of warm events (2.5 %); iii) a slightly higher soil temperature at 5 and 15 cm depths (difference of 0.1 and 0.3 °C respectively) than in case of warm events. Though it seems that there exist some little differences between the predisposing factors registered at the moment of the cold and warm glide-snow avalanche events, the differences are not statistically significant (Wilcoxon-Mann-Whitney test). The fact that in cold event, registered at the beginning of the season, soil temperatures were higher than in warm events, might be explained by the heat stored by the soil during the Summer; while for the warm events in late spring the soil was cooled by the upper snowpack and temperatures remained lower. Minor VLWC values in the basal snowpack layer during cold events represent the fact that the snow cover was cold, being formed by the first probably cold and dry snowfalls, while in spring the VLWC of the basal layer of the snow cover is the result of percolating water from above layers (the available water is more than that of cold snowpack at the beginning of the season). For the warm events at the beginning of season 2014-15, the VLWC in the snow was related to the melting of the first snowfalls occurred in November with warm air temperature (see Fig. 6 and 7).

*Gliding vs non-gliding periods*

In SA and PN the differences between gliding and non-gliding periods in terms of the measured parameters (Tab. 2 and 3) were highly significant when considering the whole dataset together, while in PN significant differences were found only in cold periods and in SA only in warm periods. Some differences could be reasonably explained with what we think the snow gliding process is, also according to existing literature (e.g. Holler, 2014 and Ancey, 2015), while others are difficult to understand. For example, in PN the VLWC in the snow was significantly higher for gliding than for non-gliding period when considering the whole dataset or cold periods. This was also found by Fromm et al. (2018) and Ceaglio et al. (2017). Instead, in SA there was no difference in the VLWC of snow during cold periods; a little significant difference was found in warm periods, but the VLWC of snow was higher in non-gliding than in gliding periods, revealing how other predisposing factors could contribute to the snow gliding process.





The multivariate CART analyses produced results, for the whole data set or dividing warm and cold periods, which show how different variables (e.g. air temperature, soil temperature and VLWC) appear in the decision process (splitting nodes) but without a general trend.

*Predisposing factors for snow gliding*

In PN for the warm gliding periods at the end of the season, the daily glide rate showed a significant ($p < 0.001$) exponential relationship with the soil VLWC at both depths (Fig. 9). Conversely, we did not find any relationship between the glide rate and the VLWC in the continuous cold gliding period in PN, as we have recently found in another experimental test site in the same region (Ceaglio et al., 2017). The difference between

those findings and the results presented here might be related also to the soil properties of the two different test sites, in particular to the Atterberg Liquid Limit. The Atterberg Liquid Limit (LL) represent the soil moisture content value determining the transition from the plastic to the liquid state (LL) (Lal and Shukla, 2004; Stanchi et al., 2012). The LL for the deeper soil horizon (10-20 cm depth) in the site of Mt de La Saxe (in Ceaglio et al., 2017) was around 48–67 %, and the registered soil VLWC was around 50 %. In PN (where we registered a cold

gliding period in 2013-14 which did not show any relationship between the glide rate and the VLWC) the LL was 53 % (topsoil) and 47 % (underlying horizon), values never measured by the WCR sensors (Fig. 4). In Mt de La Saxe soil liquefaction was possible (and actually found on field), while in PN not at all. Therefore, in Mt de La Saxe a further driving factor for snow gliding in cold periods might have been soil liquefaction, which did not occur in PN in 2014.

For SA we did not found any relationship between the daily glide rate and the VLWC for the identified both warm and cold periods.

**4 Conclusions**

In this study, through a 2 years field campaign, we analysed the predisposing factors for snow gliding and glide snow avalanches.

From the data registered in two monitoring seasons at two experimental test sites in Aosta Valley, even if the difference is not significant, we found that cold glide-snow avalanche events occurred with a higher VLWC in the soil and a lower VLWC in the basal snowpack layer than in warm events.

Significant differences were instead found between the predisposing factors during gliding and non-gliding periods, considering the whole dataset together or dividing cold and warm gliding periods. However, no general

trend during cold or warm periods was found in the two test sites.

Our analysis on the potential driving factors for gliding processes underlined the importance of VLWC, in particular for warm glide events. Though we could not find a clear and generalizable trend, the example of the warm gliding periods in 2014 in the site Pista Nera shows how the glide-snow rate increased exponentially to the VLWC in the soil.

In conclusion, among the parameters considered in this study, which sometimes showed contrasting effects, our findings contribute to assess the importance of soil water content in snow gliding processes. Therefore, this result supports the need, already suggested by other scientists, of analysing such processes with an interdisciplinary approach which integrates snow and soil science.





An effort which should be done by scientists would be to share all the data collected in the different experimental test sites, in order to create a common rich database and to be able to analyse the driving factors for snow gliding processes, including also site characteristics. Doing so, the results obtained in the different test sites might be generalized.

**Acknowledgements**

The work was developed within the Unità di ricerca "Mountain Risk Research Team" and supported by the NextData LTER Mountain project. We would like to thank the *Ufficio Neve e Valanghe* (Fondazione Montagna sicura) and the *Centro Funzionale* of the Aosta Valley Region for avalanche, snow and weather data. We also thank Laura Dublanc, Elisabetta Ceaglio, Davide Viglietti, Giuseppe Comola, Arnoldo Welf and Monterosa
S.p.a. for technical and logistic support in the experimental test sites.

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




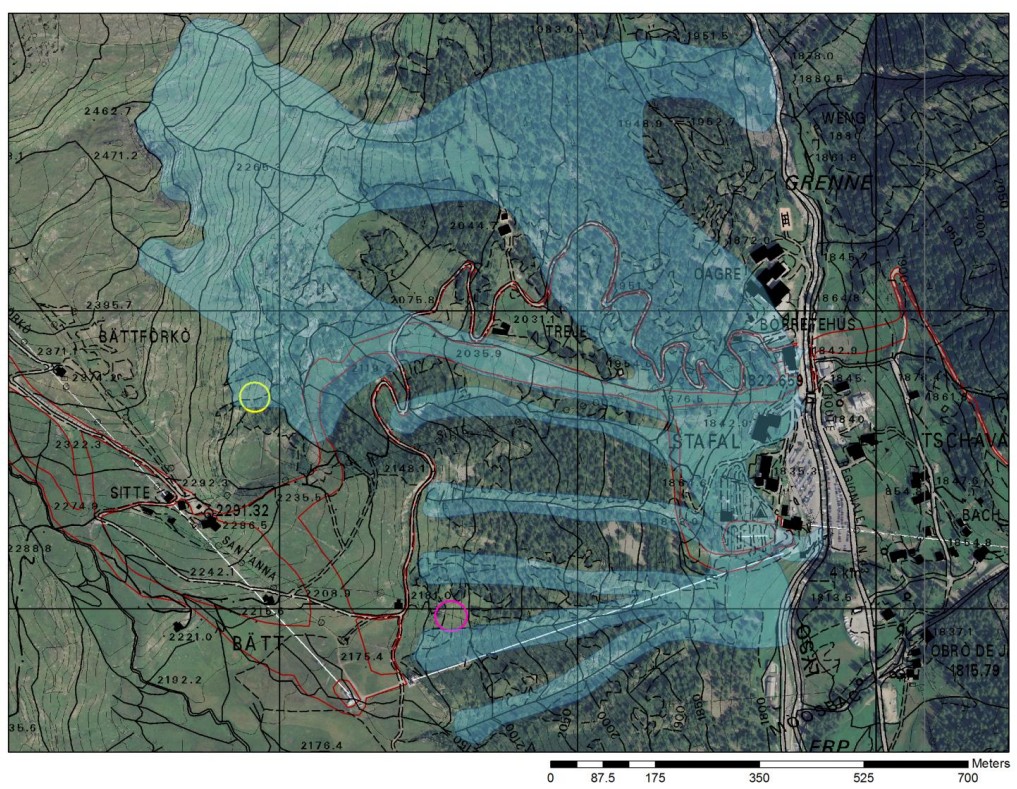

**Figure 1: Study area: location of the two test sites "Pista Nera" (PN) and "Sant'Anna" (SA), shown with the yellow and pink circles. In red the ski runs of the MonterosaSki resort are reported, while the light blue polygons represent the perimeter of the maximum events registered in the Avalanche Cadastre of the**
5 **Aosta Valley Region.**



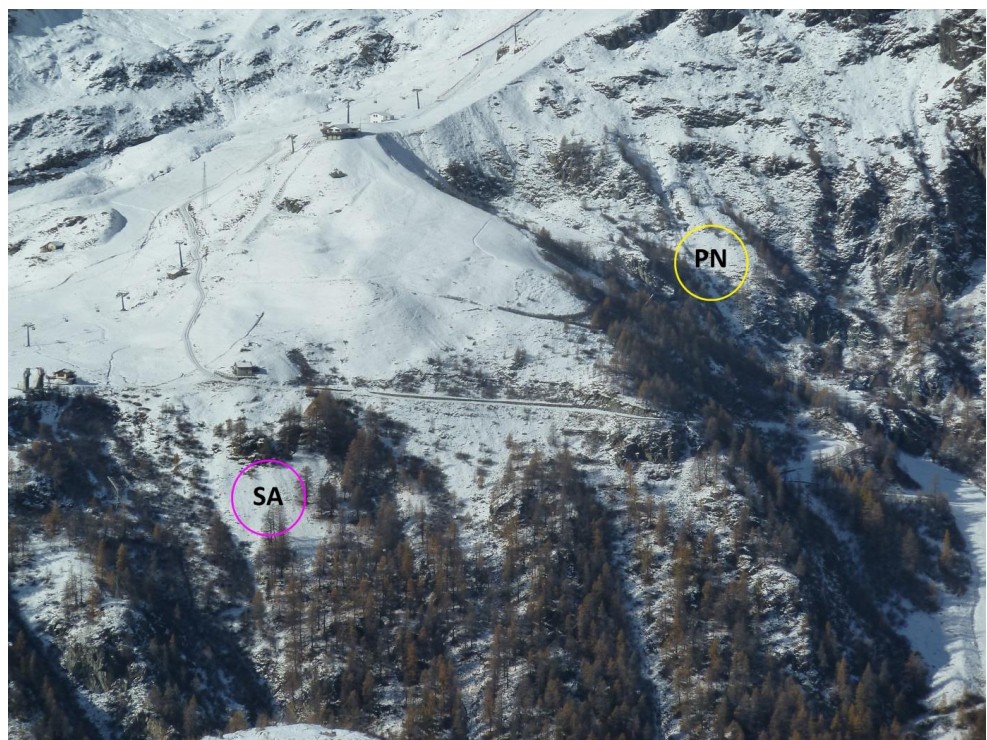

**Figure 2: Winter view of the study area: in the circles the location of the two test sites "Pista Nera" (PN) and "Sant'Anna" (SA).**




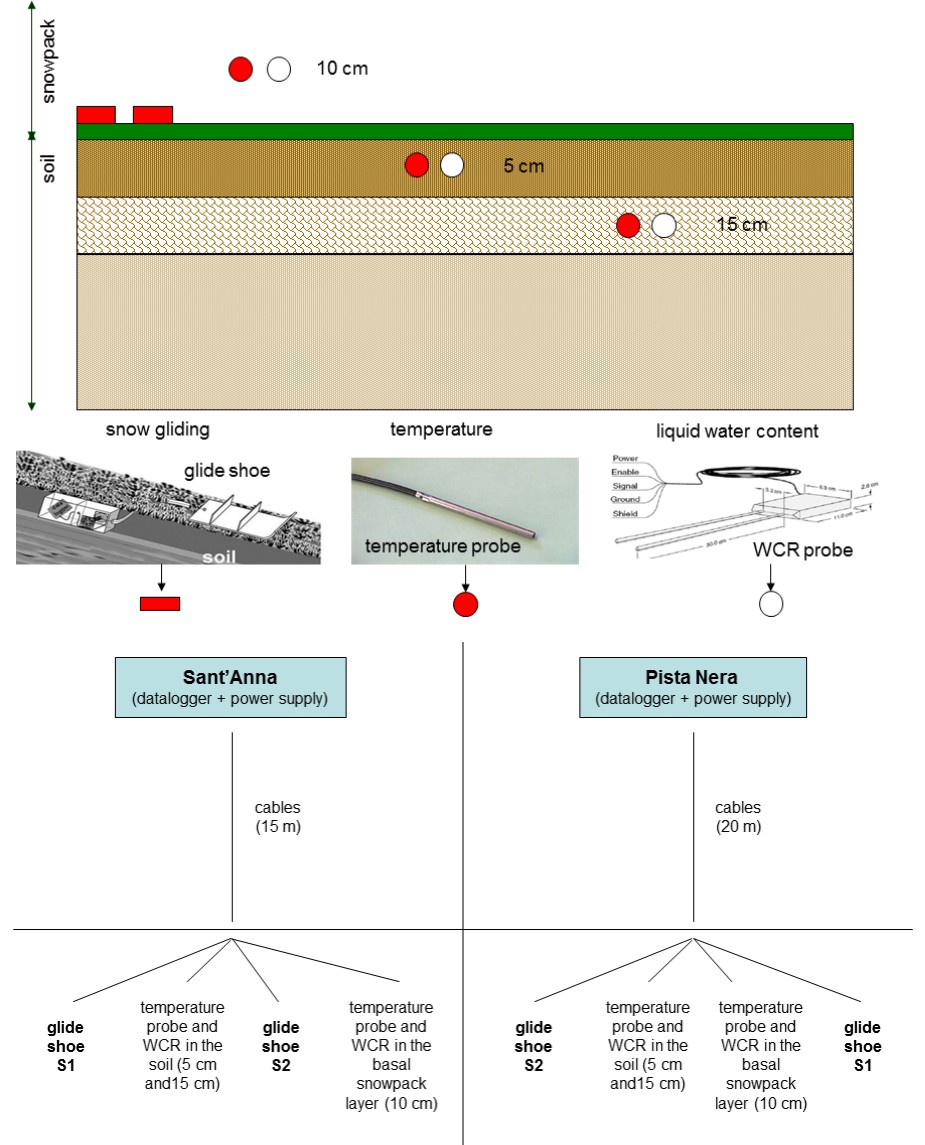

**Figure 3: Set-up of the experimental test sites.**



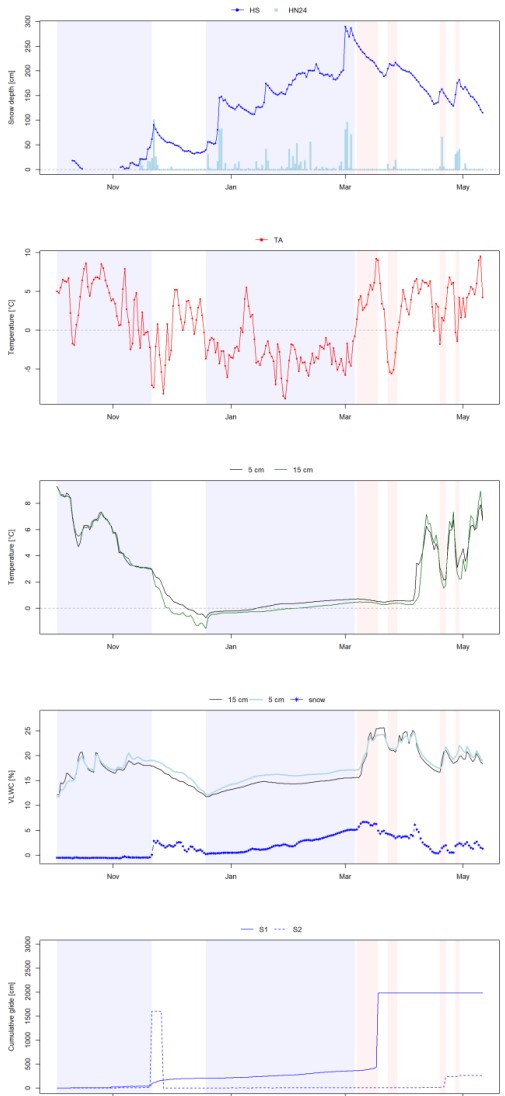

**Figure 4: Site PN, winter season 2013–2014: (a) Snow depth (HS) from AWS Weissmatten and new snow sum (HN24) from manual snow measuring site Gabiet; (b) air temperature (TA) from AWS Weissmatten; (c) soil temperature; (d) VLWC measured within the soil and in the basal snowpack layer; (e) cumulative glide. Warm and cold periods are highlighted in orange and blue, respectively; periods reported in white are periods of no data analyses.**





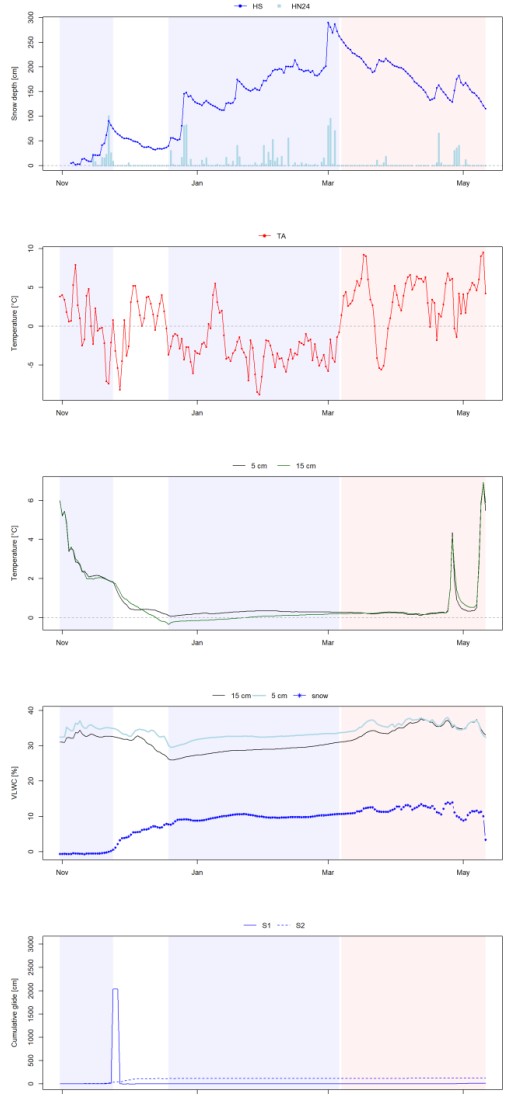

**Figure 5: Site SA, winter season 2013–2014: (a) Snow depth (HS) from AWS Weissmatten and new snow sum (HN24) from manual snow measuring site Gabiet; (b) air temperature (TA) from AWS Weissmatten; (c) soil temperature; (d) VLWC measured within the soil and in the basal snowpack layer; (e) cumulative glide. Warm and cold periods are highlighted in orange and blue, respectively; periods reported in white are periods of no data analyses.**




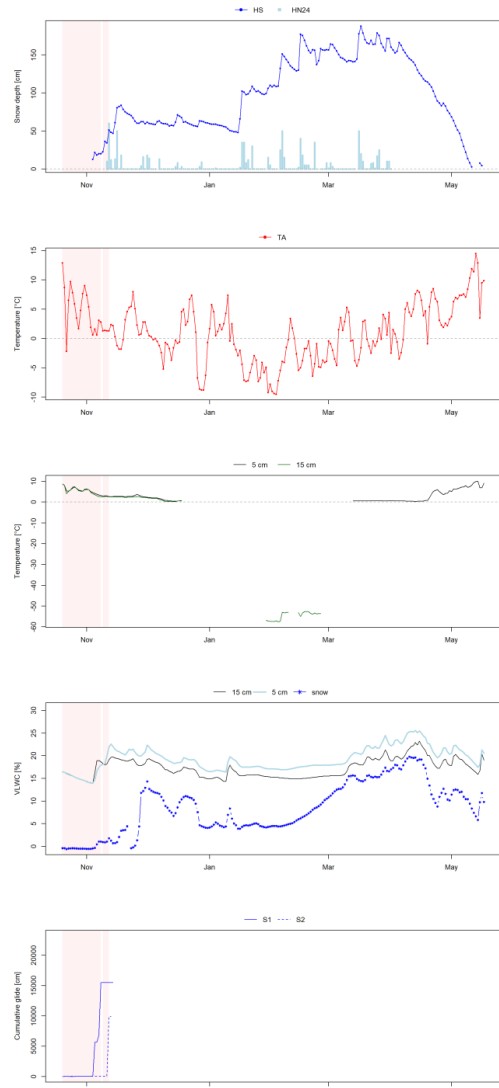

**Figure 6: Site PN, winter season 2014–2015: (a) Snow depth (HS) from AWS Weissmatten and new snow sum (HN24) from manual snow measuring site Gabiet; (b) air temperature (TA) from AWS Weissmatten; (c) soil temperature; (d) VLWC measured within the soil and in the basal snowpack layer; (e) cumulative glide. Warm and cold periods are highlighted in orange and blue, respectively; periods reported in white are periods of no data analyses (the period 9-12.11.2014 is no analyses for S1).**




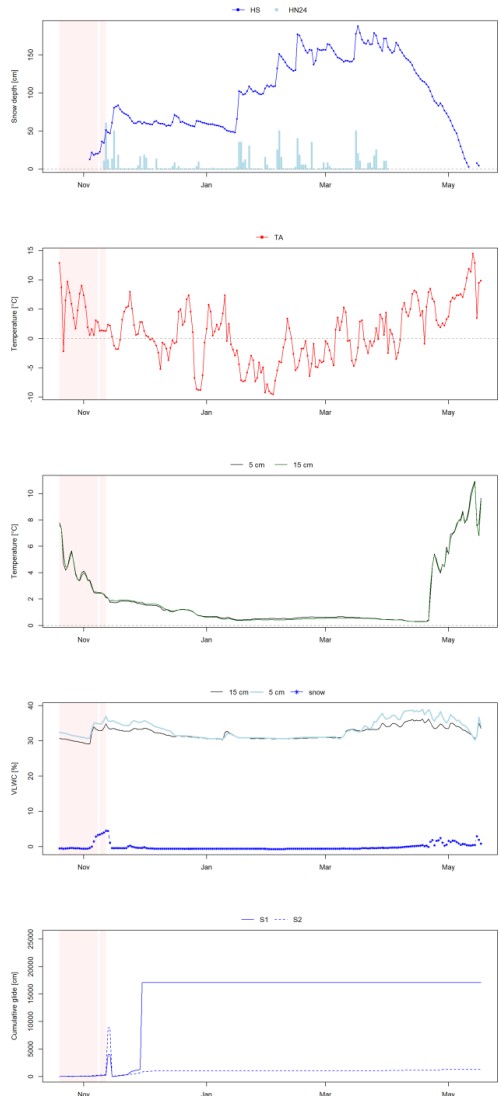

**Figure 7: Site SA, winter season 2014–2015: (a) Snow depth (HS) from AWS Weissmatten and new snow sum (HN24) from manual snow measuring site Gabiet; (b) air temperature (TA) from AWS Weissmatten; (c) soil temperature; (d) VLWC measured within the soil and in the basal snowpack layer; (e) cumulative glide. Warm and cold periods are highlighted in orange and blue, respectively; periods reported in white are periods of no data analyses (from 1.12.2014 to the end of the season no analyses for S1).**




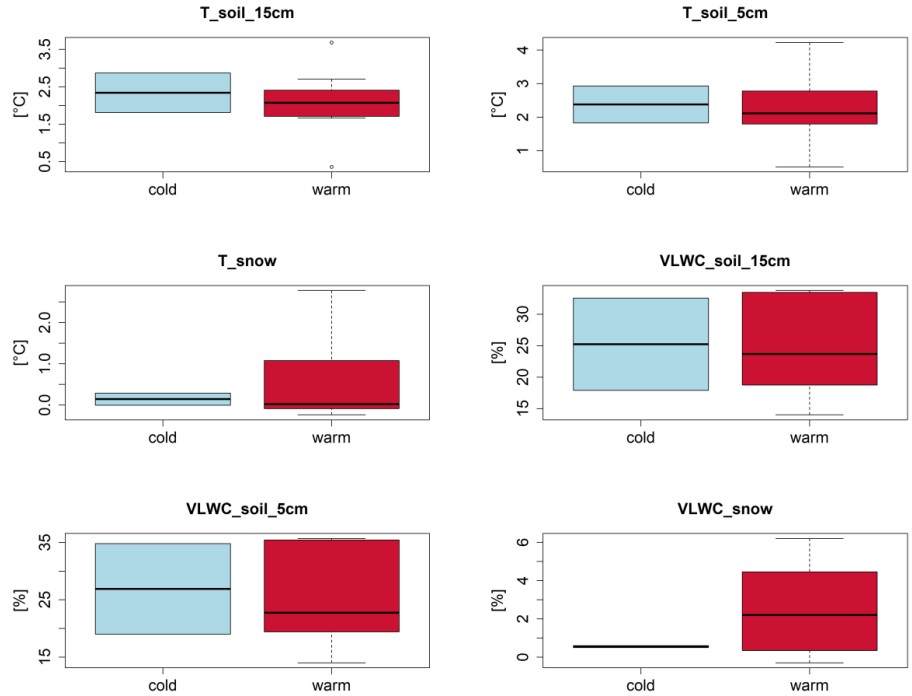

**Figure 8: Boxplots of the parameters registered at the moment of the cold and warm glide snow avalanche events (values registered 30 min before the events).**





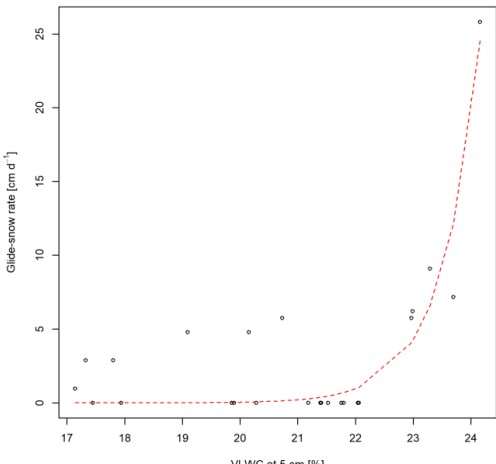

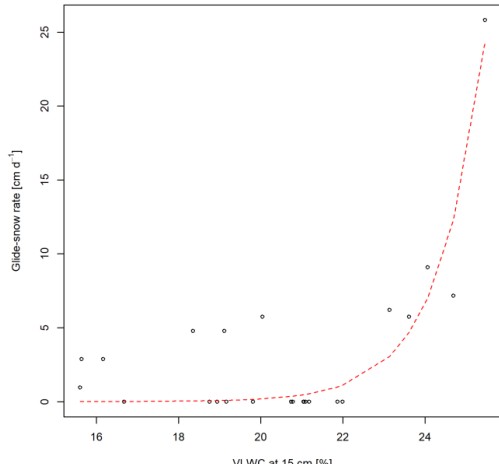

**Figure 9: Fitting model between daily glide-snow rates for S1 and volumetric liquid water content (VLWC) measured in Pista Nera at 5 cm (left) and 15 cm (right) soil depths during the warm-temperature events of season 2014.**



**Table 1. Periods of analyses for each single glide shoe in the two test sites during the two monitoring seasons. Dates are in day/month/year; N indicates the number of data in the corresponding period.**

| Period | Glide-snow shoe | Cold/warm | N |
|---|---|---|---|
| PISTA NERA | | | |
| 31/10 – 21/11/2013 | S1 and S2 | cold | 22 |
| 19/12/2013 – 6/03/2014 | S1 and S2 | cold | 78 |
| 7 – 18/03/2014 | S1 and S2 | warm | 12 |
| 23 – 28/03/2014 | S1 and S2 | warm | 6 |
| 19 – 22/04/2014 | S1 and S2 | warm | 4 |
| 27 – 29/04/2014 | S1 and S2 | warm | 3 |
| 28/10 – 8/11/2014 | S1 and S2 | warm | 12 |
| 9 – 12/11/2014 | S2 | warm | 4 |
| SANT'ANNA | | | |
| 31/10 – 24/11/2013 | S1 and S2 | cold | 25 |
| 19/12/2013 – 6/03/2014 | S1 and S2 | cold | 78 |
| 7/03 – 09/05/2014 | S1 and S2 | warm | 64 |
| 28/10 – 13/11/2014 | S1 and S2 | warm | 17 |
| 15 – 30/11/2014 | S1 and S2 | warm | 16 |
| 1 – 24/12/2014 | S2 | warm | 24 |
| 25/12/2014 – 6/03/2015 | S2 | cold | 72 |
| 7/03 – 22/04/2015 | S2 | warm | 47 |



**Table 2. Pista Nera: summary statistics showing median values of various variables for gliding days (Gd) and non-gliding days (NonGd). For each variable, distributions were contrasted (Mann-Witney U test), and the level of significance p is given. Temperatures are in °C, VLWC are in %, snow height are in cm.**

| Variables | All data | | | Cold periods | | | Warm periods | | |
|---|---|---|---|---|---|---|---|---|---|
| | Gd | NonGd | p-Value | Gd | NonGd | p-Value | Gd | NonGd | p-Value |
| T_snow | -0.19 | -0.22 | 0.173 | -0.2 | -0.2 | 0.684 | 0.75 | -0.22 | 0.354 |
| VLWC_snow | 1.8 | 0 | <0.001*** | 1.81 | 0 | <0.001*** | 1.8 | 3.57 | 0.585 |
| T_soil5cm | 0.55 | 3.13 | 0.022* | 0.37 | 3.15 | 0.043* | 2.7 | 0.58 | 0.288 |
| VLWC_soil5cm | 16.3 | 19.12 | 0.085 | 16.18 | 17.67 | 0.06 | 18.29 | 21.29 | 0.727 |
| T_soil15cm | 0.23 | 3.11 | 0.007** | -0.01 | 3.2 | 0.043* | 2.21 | 0.38 | 0.202 |
| VLWC_soil15cm | 14.8 | 18.11 | 0.04* | 14.65 | 17.26 | 0.048* | 18.75 | 20.89 | 0.601 |
| TA | -1.9 | -0.8 | 0.291 | -2.9 | -0.3 | 0.001** | 2.8 | -2.15 | 0.009** |
| HN24 | 0 | 2.5 | 0.838 | 2 | 0 | 0.805 | 0 | 5 | 0.474 |
| HS | 157 | 56 | 0.084 | 156 | 21 | <0.001*** | 163 | 212 | 0.394 |

**Table 3. Sant'Anna: summary statistics showing median values of various variables for gliding days (Gd) and non-gliding days (NonGd). For each variable, distributions were contrasted (Mann-Witney U test), and the level of significance p is given. Temperatures are in °C, VLWC are in %, snow height are in cm.**

| Variables | All data | | | Cold periods | | | Warm periods | | |
|---|---|---|---|---|---|---|---|---|---|
| | Gd | NonGd | p-Value | Gd | NonGd | p-Value | Gd | NonGd | p-Value |
| T_snow | -0.05 | -0.07 | <0.001*** | -0.05 | -0.06 | 0.619 | -0.05 | -0.08 | <0.001*** |
| VLWC_snow | -0.27 | 4.5 | 0.378 | -0.41 | -0.51 | 0.846 | -0.25 | 10.8 | 0.026* |
| T_soil5cm | 0.6 | 0.38 | <0.001*** | 0.53 | 0.44 | 0.1 | 0.64 | 0.31 | <0.001*** |
| VLWC_soil5cm | 34.58 | 32.67 | <0.001*** | 32.45 | 32.01 | 0.115 | 35.33 | 35.36 | 0.676 |
| T_soil15cm | 0.55 | 0.37 | <0.001*** | 0.46 | 0.38 | 0.139 | 0.93 | 0.29 | <0.001*** |
| VLWC_soil15cm | 32.86 | 30.88 | <0.001*** | 30.75 | 30.5 | 0.066 | 33.52 | 33.53 | 0.378 |
| TA | -0.1 | -0.65 | 0.593 | -3.5 | -2.6 | 0.361 | 1.35 | 3.6 | 0.002** |
| HN24 | 0 | 0 | 0.044* | 0 | 0 | 0.568 | 0 | 0 | 0.001** |
| HS | 127 | 144 | 0.001** | 123 | 130 | 0.164 | 139 | 156 | <0.001*** |