# Peer review of "Snow gliding and glide snow avalanches: recent outcomes from two experimental test sites in Aosta Valley (NW Italian Alps)"

_Natural Hazards and Earth System Sciences, 2019_

## Referee Comment (RC1) · Christophe Ancey (Referee) · 4 Jun 2019

This is a short paper that reports on two years of observation of snow gliding during a field survey conducted near Gressoney (Italian Alps) from winter 2013-14 to 2014-15. Snow gliding and glide avalanches are not a new threat, but the striking increase in the frequency of damage they cause has pushed scientists to further investigate the underlying physical processes, while for engineers and technicians in charge of ski resorts and high-elevation infrastructures, they have been a growing source of concern. Just for this season, a glide avalanche occurred in Crans Montana (Switzerland), killing one patrolman (but the death toll could have been more significant).

[Figure]

This paper is interesting. After reading it, I was wondering why the authors did not continue their field investigations over a longer time period, and why they submitted this paper four years later. While the authors' findings do not lead to a more accurate picture of the physical processes, the site comparison shows disparate behaviours and the diversity of explanatory factors. An interesting observation concerns the part played by the interface between the ground and snow cover. When the ground's upper layers are saturated with liquid water, then ground resistance (to snow creeping and gliding) is certainly much lower. As the authors state, little is known about this process. Empirically, in a few ski resorts, the ground is drained, with a positive influence on gliding rates. This has been, for instance, attempted in Saint-François-Longchamp (Savoie, France), with success.

On the whole, I have little to criticize. In absolute terms, I would have preferred a longer paper presenting several years of monitoring, but this may have not been possible. So, a short paper is better than no paper at all. Here are my remarks, mostly of editorial nature: 1) The paper is fairly well written, but in places, the English has to be corrected and polished. I had to read sentences several times to understand what the authors meant (e.g., p. 4, L38-39, "not like replications" -> the phrasing sounds a bit strange and unnecessary as it says the same thing as "behaved differently"). Use of acronyms should be limited to standard abbreviations. "Pista Nera" is easier to read than PN. 2) Page 2: although liquid water is present in most documented cases of snow gliding, it happens sometimes that dry snow covers also glide. This has been, for instance, observed by Damien Margueritat in Chamonix/Le Brévent (Margueritat, D., Retour d'observations sur les plaques de reptation, Neige & Avalanches, 152, 18-22, 2016). On the same page, I am not sure that the partitioning into cold and warm events is sound. On the one hand, these adjectives are borrowed from common language (boiling water is warm, ice is cold), and scientifically they do not tell us much. On the other hand, snow gliding arises from multiple scenarios, and to a large extent is history dependent. I urge the authors to be more specific in their introduction. 3) Figures 4 to 7 are very difficult to read (use larger font sizes). The same holds for Fig. 9. The paper

could be accepted after some minor adjustment of the text and figures.

Christophe Ancey

---

## Referee Comment (RC2) · Anonymous Referee #2 · 12 Jun 2019

General comments: The manuscript analyzes snow gliding processes on two experimental sites in the Aosta valley and describes how soil water content and soil temperature affects the formation of glide avalanches. The investigations may be a substantial contribution to improve our understanding about the major driving factors for snow gliding processes. However, there are some ambiguities (especially in the context of data gathering and measurement methods) which should be considered prior to publication. I recommend major revisions.

Specific points: Page2, line 11: The terms 'cold temperature events' and 'warm temperature events' were defined by Clarke and McClung (1999); thus the appropriate

citation should be included at this point.

Page2, line 26: The authors should indicate the location of the weather station D' Ejola.

Page2, line 28: On line 28 the authors used the term 'cumulative snow'; what is the meaning of this term (sum of new snow?)

Page3, line 4: The authors should indicate the location of the weather station 'Gressoney-S.J. - Weissmatten'; however, it is doubtful, if these data (the station is 12km away from the study site) are representative for the experimental site.

Page 3, line 6: The authors should indicate the location of the weather station 'Gressoney-S.J. - Lake Gabiet'; as snow distribution depends on the site (and is subject to strong variations) it is doubtful to use the new snow amount from Lake Gabiet which is 4 km away from the study site.

Page 3, line 7: The abovementioned remark is valid in particular for the application of snow profiles, which were taken from site Sant' Anna (1 km away from the study site).

Page 3, line 11: As gliding is subject to strong variations it is probably not enough to have only two glide shoes per study area.

Page 3, line 14: The authors note, that they used Campbell Water Content Reflectometers for measuring liquid water content. According to figure 3 the device was also used for measurements in the snowpack (10 cm above ground). I doubt that the device is suitable to measure water content in the snowpack, since it was especially designed for measurements in the soil.

Page 3, line 20: The authors note, that the periods for the analysis were chosen according to the specific snow situations, but these situations were obviously not gathered on the relevant study sites (as indicated in chapter 2.2. they come from the surrounding weather stations).

Page 3, line 23: How was the instrumentation reset?

Page 4, line 18: What is the meaning of a ROS event? Is it a rain-on-snow event?

Page 4, line 23: The statement that 'the warm temperature events often occurred after a snowfall followed by a sharp increase in air temperature, resulting in a decrease in the snow depth' cannot be figured out from Figures 4-7. According to Figure 6 there was a strong increase in gliding in autumn 2014. However, that rise was clearly before reaching the maximum snow depth and there was no sharp increase in air temperature.

Page 4, line 30: The authors note that in autumn 2013 snow gliding occurred earlier in PN than in SA. In order to identify these findings in the relevant Figures, it is necessary that the x-axis in Figures 4 and 5 have the same scale.

Page 4, line 38: Feistl et al. (2014) is missing in the References (I think it should be 'Feistl, Bebi, Dreier, Hanewinkel, Bartelt: Quantification of basal friction for technical and silvicultural glide-snow avalanche mitigation measures').

Page 5, line 7: The authors note that there was no snow at the ground between the event on Nov. 24th and Jan. 14th. There are two points which should be explained by the authors: 1) according to line 1 (page 5) the relevant event occurred on 21st of November and not on the 24th. 2) according to line 10 (page 5) there was a snowfall on Dec. 19th; so why was the site free of snow from November 2013 until January 14th.

Page 5, line 10: Wherefrom is the number of 30 cm of new snow, when there are no snow measurements in close vicinity to the study site (according to chapter 2 the snow measurements come from the 'Lake Gabiet' station which is approximately 4 km away from the experimental site).

Page 5, line 22: What do the authors mean with the sentence '. . .the soil was cooled by the upper snowpack'. A mighty snowpack provided, soil surface temperatures should remain more or less stable (near to 0°C) over the whole winter season.

Page 6, line 22: The Discussion is missing.

[Figure]

Page 6, line 38: A cooperation of snow and soil scientist was proposed already by Höller (2014); thus the appropriate citation should be included at this point.

References: Feistl et al. (2014) is missing (I think it should be 'Feistl, Bebi, Dreier, Hanewinkel, Bartelt: Quantification of basal friction for technical and silvicultural glide-snow avalanche mitigation measures').

Figure 3: According to Fig. 3 the Campbell Water Content Reflectometers were also used in the basal snowpack (10 cm above ground). I doubt that these sensors are suitable to measure LWC in the snowpack, since they were designed in particular for measurements in the soil.

Figures 4 – 5: To enhance comparability the x-axis in Figures 4 and 5 should have the same scale.

Figures 4 – 7: The registered 9 glide snow avalanches should be indicated by additional arrows. Thus it would be easier to compare the snow glide measurements with the relevant events.

Figures 4 – 7: The glide measurements indicate extraordinary values (about 2000 cm (20 m) in Fig. 4 and 5) and up to 15000 cm (150 m) in Fig. 6 and 7. I am wondering if such values can be measured with the Sommer-device, since the length of the wire is only 4 m in total.

Figure 6: According to Fig.6 the liquid water content in the snowpack is approximately 20

Table 1: The specified periods in Tab. 1 do not correspond with the indicated cold and warm periods in Fig. 4 – 7. The last column of Tab. 1 shows the number of data (N), but it is not clear what these numbers means in the context of snow gliding.

---

## Author Response (AR1)

Dear Editor,
In this PDF file I merge the following: 1) my response to the reviewers and 2) the changes in the revised version of the manuscript.

5 **Response to C. Ancey**

Dear Christophe, thanks for your suggestions..
We answer step by step in the following… the text is actually the comments which we have inserted directly in your PDF file through the comment tool. You find this file in the discussion.

10 • To answer to your question about the short period for monitoring the test sites... it was a founding issue. And actually also the delay in time was related to that and the involved persons. The research was planned within the MRRTeam project, but after that, unfortunately, we were not supported again for continuing the research.
In fact, we think that longer monitoring would produce more data to deepen the research questions.
15 Moreover, we think that a collaboration between researchers with test sites all over the mountains would produce results which consider all the different driving factors (vegetation included).

• Empirically, in a few ski resorts, the ground is drained, with a positive influence on gliding rates. This has been, for instance, attempted in Saint-François-Longchamp
20 (Savoie, France), with success. → Interesting as managing practice... does exist a reference about this experience? I did not find anything on the web.

• We have tried to improve the English... if not good enough yet, we will pay a native speaker to improve it.

25 • Use of acronyms should be limited to standard abbreviations. → Ok, we now limit the use of acronymous, if not standard.

• Page 2: although liquid water is present in most documented cases of snow gliding, it happens sometimes that dry snow covers also glide. This has been, for instance,
30 observed by Damien Margueritat in Chamonix/Le Brévent (Margueritat, D., Retour d'observations sur les plaques de reptation, Neige & Avalanches, 152, 18-22, 2016). → A dry snowpack can surely glide, but, to do that, liquid water should be present at its base. This is the definition of a cold event and I believe that to glide there should be liquid water at the snow soil interface... It was interesting to read the paper you suggested where you found two cases of gliding in case of cold
35 snowpack until the base of the snow cover.
I insert this citation in the Introduction...

• On the one hand, these adjectives are borrowed from common language (boiling water is warm, ice is cold), and scientifically they do not tell us much. On the other hand, snow
40 gliding arises from multiple scenarios, and to a large extent is history dependent. I urge the authors to be more specific in their introduction. → FROM THE TYPICAL AVALANCHE PROBLEMS (EAWS, 2018): "...Glide snow avalanches can occur both with a cold dry snowpack and with a warm wet snowpack. ..." Other authors (Dreier et al., 2016; Clarke and McClung, 1999) speak about warm and cold-temperature events. Maybe the term "temperature" should always be written instead of
45 using only warm and cold...... Fromm et al. (2018) divided into periods I and II, which indicate periods of rising and declining snowpack. We think it is interesting to check driving factors for those two scenarios.

• We've improved the figures, which we've changed also according to the suggestions of the other reviewer.

50

**Response to anonymous reviewer #2**

Many thanks for the very useful suggestions to improve our manuscript. I will answer point by point in the following for the major comments, while minor revisions are included in the revised version of the manuscript. The text is actually the comments which we have inserted directly in your PDF file through the comment tool.

5  You find this file in the discussion.

Page2, line 11: We've inserted this reference.

Page2, line 26: Ok, we've produced a figure to show the position of the automatic and manual measuring

10  stations for the snow and weather parameters. Doing so, we realized that the official name of D'Ejola is actually Gressoney-L.T.

Page2, line 28: We actually wrote "cumulative snowfall".... which, yes, it is the sum of new snow.

Page3, line 4: Ok, we've produced a figure to show the position of the automatic and manual measuring stations for the snow and weather parameters.

15  We consider these stations representative of the study site. Gabiet in particular is just on the opposite side of the valley at little higher altitude.

Weissmatten is 12 km south but, from previous analyses, we realized how the time evolution of the snow depth for our sites is similar to what measured at Weissmatten.

What probably is not clear from the previous version of the paper is that we had information about the conditions

20  of the sites as we collaborated with the Director of the ski-run, who weekly, even daily sometimes, report us the conditions of the area.

This comment is valid also for the following remark (Page 3, line 6).

Page 3, line 7: We used the snow profiles made weekly at the official measuring site of the Regione Valle d'Aosta to obtain existing periodical data from already existing measuring sites. We think these data are useful to

25  follow the seasonal evolution of the snowpack in the area. However, we also made ad-hoc snow profiles at the sites, especially after some specific gliding events.

We add a sentence regarding these specific snow profiles in the manuscript.

Page 3, line 11: We did not have in mind to use two glide shoes as replications. In fact, the glide shoes in the same test site behaved differently... we discuss this fact in the Results and Discussion section.

30  Our sites were not dedicated only to gliding (for which we would have used a higher number of glide shoes, according of course to the available budget, to make replications - as for example in some Austrian test sites) but aimed also at catching the glide avalanches, which can interfeer with both or only one glide shoe.

Page 3, line 14: Yes, thanks!

Actually we know this and in fact, parallel to this paper, we made a specific campaign to find a way to correlate

35  the values measured by the WCR in the snow with values determined starting from snow density and Denoth equation. We forgot to mention this work. We add now a sentence to tell that we corrected the value registered by the probe placed in the basal snowpack layers following the corrections found by Godio et al (2018) during a specific experimental campaign performed in the vicinity of the manual measuring site Sant'Anna.

Page 3, line 20: Actually it was not clear from the text (that we have changed according to your suggestion)

40  that we also went to the site for specific field work, for example in case of significant gliding events and for resetting the instrumentation.

In such occasions we made specific observations which helped us to choose the periods of anlayses.

Moreover, also the Director of the ski-run gave us information on the specific local conditions.

For example, after a glide avalanche, the site was free of snow therefore the time from the glide avalanche and

45  the following snowfall was not considered in the analyses as the site were snow free.

Page 3, line 23: We went into the field to recover the glide shoes involved in the events and to replace them close to the potenziometer, which we also checked. We add a sentence at the end of the Data collection section.

Page 4, line 18: yes!

Page 4, line 23: You are right, better not to generalize.

50  We've changed the sentence and highlight better the event when this fact is clearly visible.

Page 4, line 30: we made it!

Page 4, line 38: we've add the missing reference.

Page 5, line 7: 1) Thanks. We have changed 24 to 21.

2) We've changed a bit the text to be clearer. We've actually not written that the site was free of snow. S2 was

55  free of snow just after the event on November 21st or covered by little snow.

Page 5, line 10: See the previous comments about AWS and snow measuring sites.

This value come from the manual measuring site Lake Gabiet, which is representative of the site.

Also the Director told us it was snowing again.....

**Page 5, line 22:** You are right, the sentence is confusing. It is not a proper cooling but an impediment for the soil to warm up. We've changed the sentence.

**Page 6, line 22:** Yes, we have decided to merge Results and discussions.

We have asked the Editor, who also in the pre-discussion process told us to separate the sessions. But we made a version with separate sections which we liked less.... we think it comes natural to discuss already the results. In this way we also avoid repetitions. We've discussed with the Editor and finally he let us free to decide if merging or not.

**Page 6, line 38:** Yes, we know!

We've in fact cited that work in Ceaglio et al. (2016):

*Höller (2014) concludes that "The increasing number of glide snow avalanches in certain winter periods might be associated with the soil and ground surface conditions in late autumn and early winter; however, this assumption is primarily based on observations and not yet confirmed by relevant investigations. In this context, the soil conditions and the conditions at the snow–soil interface should be investigated."*

But, it is true, we should do it also here. We've cited that work.

Concerning the Figures, we have changed them accordingly to all your suggestions.

**Figures 4 – 7: The glide measurements indicate extraordinary values (about 2000 cm (20 m) in Fig. 4 and 5) and up to 15000 cm (150 m) in Fig. 6 and 7. I am wondering if such values can be measured with the Sommer-device, since the length of the wire is only 4 m in total.** → Concerning the measuring device by Sommer, the latest version has a cable of 20 m! But MANY THANKS: in Fig. 6 and 7 there is a "0" too much......... the correct values were an order of magnitude lower. We corrected the graphs of the cumulative glide.

**Figure 6: According to Fig.6 the liquid water content in the snowpack is approximately 20.** → Yes, we realized that this value is much too high. However, the period in white is a period of no data analyses, therefore these values did not affect our results.

I explain to you what happened...

After the event of November 2014, we could not reset the glide shoes due to unsafety conditions in both sites. Therefore, we also could not check the position of the WCR probe. Therefore such high values, close to values registered in the soil, might be explained by the fact that the WCR probe moved during the avalanche and finally was accidentally inserted in the soil.

Not to make confusion we could eventually cancel the data after the event on November 2014...

But, then, we should do that for all graphs....

I would leave as it is now, stressing the fact that white periods represent periods of no analyses when the recorded values at the two sites might be unrealistic.

We add a sentence about the unrealistic values from VLWC (and eventually other sensors) in periods of no analyses in the Data analysis section.

**Table 1:** THANKS! you are right!

In particular for the beginning of the seasons and in Fig. 7 for S2 for which the period of analyses continued also after the event in November 2014.

The number of data (N) are the number of the days when the analyses have been performed. As we analized daily rate N gives an indication of the data we could use in the analyses. N does not represent anything in term of snow gliding, it is not the number of the days when gliding was registered. We just wanted to give a number to show how much data we have for statistics.

Concerning the warm and cold periods in the figures and in the table: in the Table we can give more details as we can report the different periods for S1 and S2, while in the figure we've drawn an indication of warm and cold periods which cannot take into account this distinction... however we think it is useful as overview on warm and cold periods.

[revised manuscript text omitted]